# Effect of Diet and Exercise-Induced Weight Loss among Metabolically Healthy and Metabolically Unhealthy Obese Children and Adolescents

**DOI:** 10.3390/ijerph19106120

**Published:** 2022-05-18

**Authors:** Qin Yang, Kun Wang, Qianqian Tian, Jian Zhang, Linyu Qi, Tao Chen

**Affiliations:** 1International College of Football, Tongji University, Shanghai 200092, China; yangqin@tongji.edu.cn (Q.Y.); 2132026@tongji.edu.cn (J.Z.); 2055021@tongji.edu.cn (L.Q.); 2Shanghai Dianfeng Sports Management Co., Ltd., Shanghai 200441, China; 3Shanghai Jiao Tong University School of Medicine, Shanghai 200025, China; kittytian@shsmu.edu.cn; 4China Hospital Management Institute, Shanghai Jiao Tong University, Shanghai 200025, China; 5Sports and Health Research Center, Department of Physical Education, Tongji University, Shanghai 200092, China

**Keywords:** obesity, children and adolescents, metabolically healthy obesity, weight loss, diet, exercise

## Abstract

Objective: To study the effect of diet- and exercise-based lifestyle intervention on weight loss (WL) and cardiovascular risk among metabolically healthy obese (MHO) and metabolically unhealthy obese (MUO) children and adolescents. Methods: The sample included 282 obese individuals (54% males, age (±SD) 12.9 (±2.3) years) who completed a 3- to 4-week WL camp program between 2017 and 2019. MUO was defined according to the consensus-based definition of pediatric MHO in 2018. Results: The intervention exhibited significantly benefits in improving body weight, body mass index, body fat ratio, waist circumference, systolic blood pressure (SBP), diastolic blood pressure (DBP), resting heart rate (RHR), triglycerides (TG), total cholesterol, and low-density lipoprotein–cholesterol levels in both MHO and MUO groups (for all comparisons, *p* < 0.01). However, the beneficial high-density lipoprotein–cholesterol (HDL-C) level (both *p* < 0.01) decreased evidently in both groups after intervention. In addition, percent changes in SBP (*p* < 0.001), DBP (*p* < 0.001), RHR (*p* = 0.025), fasting blood glucose (*p* = 0.011), and TG (*p* < 0.001) were more profound in MUO group than that in MHO group. Conclusion: Metabolical health is a mutable and transient state during childhood. Although both groups gained comparable WL benefits from diet- and exercise-based lifestyle intervention, the MUO group may benefit more than the MHO group. Strategies aiming at lowering blood pressure and preventing the decrease of HDL-C level should be considered for the precise treatment of childhood obesity in clinical practice, with the goal of improving metabolically healthy state.

## 1. Introduction

It is well established that the global epidemic of obesity, with the accompanying rise in the prevalence of endocrine and metabolic disorders, would lead to a marked increase in cardiovascular disease (CVD) [1,2]. In China, overweight and obesity have increased substantially and the latest national prevalence estimates for 2015–2019 was 11.1% for overweight and 7.9% for obesity in children and adolescents aged 6–17 years [3]. As a vital point in the life course, adolescence is characterized by rapid and transformative physical, cognitive, and emotional growth, making it a vulnerable group to unhealthy influences, such as unhealthy diets, sedentary lifestyles, and other recognized risk factors [4].

Family-based lifestyle interventions, including dietary modifications and increased physical activity [5], are sufficient to produce notable health benefits both in anthropometry and cardiometabolism, and have been considered as the cornerstone of weight management in children and adolescents [6]. However, dieting-induced metabolism adaptations in the homeostatic system that control body weight [7,8] and unsustainable behavioral habits and cognitive factors will hinder long-term weight loss effects [9]. At present, there are still great difficulties in the long-term treatment of obesity in children and adolescents due to the rising prevalence and limited healthcare resources. Thus, it is necessary to carry out refined management of obesity based on different perspectives.

In recent years, a growing interest has been raised regarding a distinct subgroup of obese individuals called “metabolically healthy obesity” (MHO), which was observed with normal blood pressure (BP), blood lipids, blood glucose, and insulin sensitivity, despite having excessive body fatness [1]. Distinguishing obesity based on metabolically healthy status will be useful to identify individuals or subgroups with high cardiovascular and metabolic risks and to optimize prevention and treatment strategies of obesity [10,11].

In adults, numerous studies have confirmed the comparable health benefits in response to diet- and exercise-based intervention between MHO and “metabolically unhealthy obesity” (MUO) groups [12,13,14,15,16,17,18], while greater weight loss may produce more cardiovascular health benefits [16,18]. However, the consensus on MHO in children was introduced only recently in 2018 [19], and few studies have focused on the effect of diet- and exercise-based intervention on children with different metabolically healthy status [1]. In fact, the metabolically healthy status in childhood is vital because many of the metabolic-related disorders begin in early childhood and over time significantly increase the CVD risk in young adults [20]. Therefore, this study evaluated the influence of a traditional diet- and exercise-based intervention on weight loss (WL) and cardio-metabolically health state among MHO and MUO children and adolescents. The results can be used to contribute to precise management and treatment of children obesity according to their metabolically health status.

## 2. Materials and Methods

### 2.1. Study Population and Database

282 obese children and adolescents (54% males, age (±SD) 12.9 (±2.3) years) who volunteered to participate in the WL Summer Camp program at the Shanghai Dian Feng Weight Loss Center between 2017 and 2019 were evaluated. It is a national standard-setting unit for weight control designed to educate overweight and obese patients to establish healthy lifestyle habits based on a “5D Weight Loss Education System” (Mindset, Goal, Approach, Motivation, and Action). The center operates all weight control services, including diagnostic testing, physical examinations, personalized diet and exercise programs, blood analysis, online guidance, educational group sessions, and follow-up services. The patients were not recommended for the WL Summer Camp if they presented concomitant renal, hepatic, cardiac disease, and/or were being treated with bariatric surgery or medications that would affect the body weight (BW) in the initial screening. A sample of 1063 participants were recruited from WL centers located in Shanghai and Beijing and included in this study. Subjects whose age were not recorded or who were ≤6 years or ≥19 years (*n* = 37) were excluded from the analyses. Individuals with missing initial or post-intervention anthropometric and metabolic parameters (*n* = 598) and those who had a body mass index (BMI) ≤ age- and sex-specific 95 percentiles (*n* = 38) and who attended the camp less than 3 weeks or over 4 weeks (*n* = 105) were also excluded. Patients with extreme values of measurements (<1 percentile or >99 percentile) (*n* = 3) were considered outliers and excluded from the analyses. Finally, 151 males and 131 females were included in this research. The detail protocol and written informed consents were obtained from all participants and their parents. The study protocol was approved by the local institutional ethics committee and adhered to the tenets of the Declaration of Helsinki (Ethics approval number: 2021tjdx046; 9 March 2021). This research is conducted independently, and the findings and conclusions of the research are not influenced by related centers.

### 2.2. Diet and Exercise Protocol

All subjects received a professional assessment and individualized diet and exercise advice for 3 to 4 weeks (Average intervention days for all participants: 24.9 ± 1.9 days, MHO: 24.7 ± 2.4 days; MUO: 25.1 ± 1.5 days, *p* = 0.180). The recommended daily energy intake was based on basal metabolic rate requirement obtained from body composition measurement during intervention [21]. The caloric intake was calculated based on the Chinese food chart and food categories recommended by Dietary Guidelines for Chinese Residents (2016) were selected [22]. The types of breakfast foods usually include buns, soy milk, milk, porridge, and eggs. The types of foods for lunch and dinner mainly include vegetables (lettuce, cabbage, celery, broccoli, radish, cauliflower, tomato, mushroom, etc.), high-protein meats (red beef and pork, fish and shrimp meat, chicken breast, etc.), eggs (mainly eggs) or beans (mainly soybeans and their products), coarse cereals (mainly rice and steamed bread) or potato foods (potatoes, sweet potatoes, yams, etc.), and a piece of after-meal fruit (usually banana, apple, orange, watermelon, cantaloupe, pitaya, orange, etc.). Three well-balanced meals were provided each day with the following calorie allocations according to previous methods [21]: protein, 20% to 30%; carbohydrate, 50% to 60%; and fat, 20%. Breakfast accounted for 35% of the total daily energy intake, lunch for 40%, and supper for 25% [21]. The prescribed diet was formulated by dedicated nutritionists, and pivotal nutrients such as vitamins, minerals, essential amino acids, fiber, and polyunsaturated fatty acids were included.

All participants applied the same incremental exercise testing as previous methods [23] on the premise that the resting 12-lead electrocardiogram was normal. Participants began the running on a flat treadmill (H/P/Cosmos Pulsar, Nussdorf-Traunstein, Germany) at a speed of 4 km/h, increased by 2 km/h every 2 min, and then paused for 10 s to record the immediate electrocardiogram (ECG) at the end of each exercise load level, followed by the next exercise load until 8 km/h was reached. The test was stopped if the subject could not bear the exercise intensity or abnormalities appeared in the ECG. The exercise program (6 days/week, 2 sessions daily, 2 h/session) consisted of jogging, aerobics, basketball, swimming, badminton, etc. Each session included a warm-up, aerobic exercise at an intensity of 20–40% of heart rate reserve (220-age-resting heart rate (RHR)), and a cool down stage [23,24]. Exercise intensity was monitored with a finger clip pulse oximeter recording heart rates when feasible. In addition, Borg’s rating of perceived exertion (RPE) was applied to assist in adjusting individual exercise intensity. Taking basketball class as an example, the 2-h section is divided into 2 classes. The first class mainly includes four parts: (1) preparatory activities (15 min, about 20% HRR, mainly jogging and dynamic stretching of joints and muscles of the whole body), (2) basic basketball skills practice (15 min, about 20–30% HRR, mainly including body posture practice, training without the ball and then practice with the ball), (3) fun games (15 min, about 30–40% HRR, mainly including fun passing, running and other mini-games), and (4) rehydration and rest (15 min). The second class also includes four parts: (1) small field games (20 min, about 40% HRR, 4 vs. 4), (2) physical fitness exercise (20 min, about 20–30% HRR, upper and lower body explosive power, strength endurance, agility, and coordination exercises, etc.), (3) post-exercise relaxation activities (10 min, mainly static stretching), and (4) discussion and communication (10 min, mainly to summarize the performance of this activity). Professional physicians and trained coaches were employed to ensure the health eligibility and safety of all participants. In addition, subjects were encouraged to develop good lifestyle habits through health lectures, nutritional and kinesiology knowledge, early bedtime and early riser, and less screen time, etc.

### 2.3. Data Collection

Questionnaire surveys were sent out to collect demographic characteristics (age, sex), medical history, and lifestyle information. Similar to measurements used in previous studies by our team [21,24], BW and height were measured using a digital scale (Yaohua Weighing System Co., Shanghai, China) and a wall-mounted stadiometer (TANITA, Tokyo, Japan) following the manufacturer’s instructions [25], respectively. BMI was calculated by BW in kg/m^2^. Waist circumference (WC) and body fat ratio (BFR) were measured using an impedance analyzer (TANITA, Tokyo, Japan), and systolic blood pressure (SBP), diastolic blood pressure (DBP), and resting heart rate (RHR) were measured using a sphygmomanometer (Nishimoto Sangyo Co., Tokyo, Japan) following the manufacturer’s instructions [21,24]. Twelve-hour overnight fasting blood samples at baseline and after-intervention were centrifuged, aliquoted, and immediately frozen and further analyzed by Adicon Medical Laboratory Center (Shanghai, China), which was certified by the China Inspection Body and Laboratory Mandatory Approval (CMA). All instruments were calibrated every day and all assessments were conducted by trained surveyors during the research.

### 2.4. Definition of MHO and MUO

According to the consensus reached in 2018 [19], a definition of MUO is as follows. Meeting one or more of the risk factor sets: (1) high-density lipoprotein–cholesterol (HDL-C) ≤ 1.03 mmol/L (or ≤40 mg/dL); (2) triglycerides (TG) > 1.7 mmol/L (or >150 mg/dL); (3) SBP or DBP > 90th percentile; (4) fasting blood glucose (FBG) > 5.6 mmol/L (or >100 mg/dL). If none of the risk factor sets were present, the patient was categorized as MHO. In addition, the latest industry standard for diagnosis of obesity (age- and sex-specific 95 percentiles) and abnormal blood pressure (age-, sex- and height-specific 90 percentiles) in Chinese children and adolescents was adopted in our study [26,27].

### 2.5. Statistical Analyses

Descriptive data are presented as mean ± standard deviations (SD) and analyzed by SPSS Statistics 25.0. Normality of all variables was tested by the Kolmogorov–Smirnov test. Independent t tests and chi-square analyses were conducted where applicable to compare the continuous variables and categorical variables at baseline, respectively. Differences of anthropometric and metabolic indicators before and after intervention in each group were analyzed via Paired t test. For comparison of percent changes in variables between MUO and MHO groups, analysis of covariance (ANCOVA) with metabolically healthy status as the between-subjects factors with inclusion of baseline level of dependent variable and other confounding variables as covariates. Interactions of sex and age with groups (MUO and MHO) were also considered. As no evidence of interactions were observed, the analysis was conducted using the whole sample. Significance was set at a 2-tailed *p* value < 0.05.

## 3. Results

### 3.1. Basic Characteristics of the Two Groups

A total of 282 obese Chinese children (mean age, 12.89 ± 2.28 y; 53.5% male; mean height, 162.41 ± 10.94 cm; mean BW, 81.65 ± 19.44 kg; mean WC, 97.4 ± 11.4 cm) were included and grouped by stratification of metabolically healthy status in this study (Table 1). Overall, 102 (49.1% males) and 180 (56.7% males) of the obese children were MHO and MUO at baseline, respectively. MHO children were relatively younger (*p* = 0.010) with lower height (*p* = 0.002), BW (*p* < 0.001), and WC (*p* < 0.001) compared with the MUO individuals.

### 3.2. Distribution of Indicators Related to Metabolically Healthy Status of the Two Groups before and after Intervention

Among MUO children, 90 (50.0%) subjects present only one risk factor, 59 (32.8%) with two risk factors, 26 (14.4%) with three risk factors, and only 5 (2.8%) had all four risk factors on the definition of the MUO phenotype (Appendix A). Table 2 shows the frequency changes of metabolically healthy indicators that were used to distinguish MHO from MUO before and after intervention. Although meeting the MHO definition criteria at baseline, 36 subjects (35.3%) in the MHO group transitioned to MUO state after intervention. Accordingly, an increased frequency of low HDL-C (28.4%), hypertension (6.9% with higher SBP and 5.9% with higher DBP) and hyperglycemia (1.0%) were found in our results. In contrast, the most frequent metabolically healthy risk factor in the MUO children was hypertension (65.0% with higher SBP and 48.9% with higher DBP), followed by low HDL-C (42.2%), hypertriglyceridemia (12.2%), and FBG (1.7%) at baseline. As expected, the frequency of hypertension (20.0% with higher SBP and 17.8% with higher DBP) and hypertriglyceridemia (1.1%) decreased remarkably except for low HDL-C (56.7%), the frequency of which displayed a moderate increase after intervention in MUO group. In addition, a total of 68 individuals transitioned from MUO state to MHO (28.9%) or non-obese status (8.9%, BMI≤ (age- and sex-specific 95th percentile)) after intervention.

### 3.3. Changes of Anthropometry and Blood Indicators in MHO and MUO Groups before and after Intervention

In the MHO group, a distinct decrease in most of the anthropometry (BW, BMI, BFR, WC, SBP, DBP, and RHR) and blood indicators (TG, TC, HDL-C, and low-density lipoprotein–cholesterol (LDL-C)) were found after intervention compared with before intervention (for all comparisons, *p* < 0.01), except for FBG, which only showed a downward trend, but did not reach statistical significance (*p* > 0.05) (Table 3). In contrast, all the above indicators improved evidently after intervention when compared with before intervention in MUO group (for all comparisons, *p* < 0.01). Analysis of covariance (ANCOVA) displayed that the percent changes of SBP (*p* < 0.001), DBP (*p* < 0.001), RHR (*p* = 0.025), FBG (*p* = 0.011), and TG (*p* < 0.001) in the MUO group were more prominent than that in the MHO group in responding to intervention, while percent changes of BW (*p* = 0.317), BMI (*p* = 0.077), BFR (*p* = 0.292), WC (*p* = 0.357), TC (*p* = 0.670), HDL-C (*p* = 0.121), and LDL-C (*p* = 0.730) were comparable between the two groups. In general, these results suggest that the MUO group may benefit more significantly than MHO group in modulating CVD-related metabolically healthy risk when dealing with diet and exercise intervention.

## 4. Discussion

To our best knowledge, this study is the first time to observe the effects of a traditional diet- and exercise-based intervention on WL- and CVD-related risk among MHO and MUO Chinese children and adolescents. Consistent with previous reports, MHO children were significantly younger and of lower excess body weight compared with MUO children [11,28], who were characterized by more frequent manifestations of hypertension, low HDL-C, and hypertriglyceridemia successively at baseline. Diet- and exercise-based intervention has a similar regulation effect on reductions in BW, BMI, BFR, WC, TC, HDL-C, and LDL-C between the two groups, while improvement of SBP, DBP, RHR, FBG, and TG in the MUO children was more prominent than that in MHO group. These results are in accordance with those reported in adult studies [14,16,17] and suggest that diet- and exercise-based intervention is quite beneficial to both MHO and MUO individuals, and the latter may benefit more. Moreover, the observation that 37.8% of MUO children (68 of 180) transitioned to MHO (52 subjects, 28.9%) or non-obese (16 subjects, 8.9%) status after intervention indicates a strong metabolic plasticity in childhood compared with adulthood [17]. Finally, the changes in HDL-C levels after intervention in both groups give us a hint that strategies should be taken to prevent HDL-C declines in diet- and exercise-based WL. Our results will provide favorable recommendations for the precision management of childhood obesity based on metabolically healthy status.

Previously reported prevalence of the MHO phenotype in children varies 20 to 68%, based on the different MHO definitions and study populations [11,29,30]. The prevalence of MHO phenotype in our study (36%) is higher than that reported by Chen F et al. (15.3%) [31], analogous with that reported by Genovesi S et al. (39%) [11], but lower than another study reported by Reinehr T et al. (49%) among large obese children populations [28]. For distribution of risk factors among MUO children, the results that 50% of the MUO children present only one risk factor and the most frequent metabolic risk factor in the MUO children is high SBP are consistent with the above reports [11,28] and indicate that most MUO phenotype in childhood and adolescence is relatively mild, and management of blood pressure, especially SBP, should be one of the important goals to protect from MUO.

For morphological indicators, although there were significant differences in baseline BW, BMI, and WC between the two groups, the authors found that metabolically healthy status has no significant effect on the percent changes in these indicators after intervention with age and sex adjusted, suggesting a considerable benefits in improving these morphological indicators (including BFR) between the two groups. These results are in accordance with previous studies [14,17] and confirmed the well-matched WL effects between MHO and MUO children.

Hypertension, an important indicator to distinguish between MUO and MHO [1], is the most frequent CVD-related risk in the MUO individuals in our study. The mechanisms of hypertension in obese children are complex and may be associated with sympathetic activation, renin-angiotensin system activation, inflammation, endothelial dysfunction, and oxidative stress [32]. WL, through healthy lifestyle modifications, such as diet and physical activity, is the cornerstone in the treatment of obesity-related hypertension [9,33]. Accordingly, decrease in BMI has been reported to be associated with decreases in blood pressure and blood lipids [34]. As expected, the authors found that diet- and exercise-based lifestyle intervention improved SBP and DBP significantly, whether in the frequency of hypertension or the range of BP changes in MUO group. Although within the normal level at baseline, a moderate but significant drop in BP after intervention in MHO group was also observed. The reason why the decrease range of BP in the MUO group was greater than that in the MHO group may be due to the distinct difference in the initial BP between the two groups. These results are in line with some of the previous studies on adults [14,16,18] and show that lifestyle intervention is beneficial to BP control in both MHO and MUO group. On the other hand, the better regulation effects of BP may be one of the reasons why lifestyle intervention brings further health benefits for individuals in MUO group, for no change in BMI was found to correlate with no change in blood pressure among children and adolescents in a recently published study [34].

The greater changes of RHR in MUO group than that in MHO groups suggest a more distinct improvement of the cardiovascular fitness after intervention [35]. Although FBG was included as one of the criteria for defining MUO [19], there was no individual who belonged to the MUO group simply for FBG > 5.6 mmol/L in our study. The three cases of hyperglycemia in MUO group were all accompanied by other risk factors that used to define MUO (Appendix A). This phenomenon was similar to the lower proportion of hyperglycemia in MUO phenotype reported in other studies [11,17]. Therefore, FBG combined with other indicators, such as insulin resistance, glucose intolerance, glycosylated hemoglobin, or insulin sensitivity [19], may be more reliable to identify MUO. The phenomenon of similar baseline FBG levels between the two groups but evidently lower (still within the normal range) in MUO group than that in MHO group after intervention indicated a greater plasticity and stability in glucose regulation in MUO individuals. In addition, the reason why improvement of TG was more profound in MUO group than that in MHO group was similar to the above-mentioned BP changes.

The similar change range of TC, HDL-C, and LDL-C in the two groups after intervention indicated a parallel effect that might be independent of metabolically healthy status. Notably, HDL-C level decreased significantly after intervention in both groups. Indeed, there is a paradoxical link between diet- and exercise-induced weight loss in children and adolescents and HDL-C levels, which has been shown as increased [36], unchanged [37], or decreased [21,38] in the concentrations of this recognized biomarker of cardiovascular health. A biological plausibility for reduction in HDL-C level may be the metabolism related to fat intake, as fatty acids are substrates for HDL-C components, especially those smaller, more dense particles exhibiting greater protective potential [38,39]. Moreover, the lower levels of HDL-C might indicate fewer particles to mediate its multiple functions. Aicher BO et al. [40] reported that the unreduced apolipoprotein A-I levels and enhanced reverse cholesterol transport by the ABCA1 transporter might facilitate the cholesterol efflux capacity of HDL-C. Therefore, the decrease in HDL-C in this setting may not be associated with increased CVD risk. In addition, it was worth noting that the decline in HDL-C level was also the main reason for the 36 MHO individuals shifting to the MUO state, suggesting that the metabolically healthy status is highly variable and therapeutic goals aimed at protecting from the decrease of HDL-C should be considered for both MHO and MUO individuals. A Mediterranean diet, especially when enriched with virgin oilseeds and olive oil [39,41], and the addition of aerobic and resistance training during WL program, will significantly enhance parameters of HDL-C cardioprotective functions [38,42,43].

This study has several limitations. First, the baseline data is not consistent, as the MHO group tends to be younger and less obese than MUO group due to the study’s retrospective design. The large age range of subjects may be the main reason for this difference. Indeed, some anthropometric and metabolic variables are very different depending on age and sex among children and adolescent groups [44,45]. Thus, future attention should be paid to age- and sex-based anthropometric and metabolic changes in these groups during weight loss to obtain more convincing results. Second, the authors focused on a specific population of MHO/MUO children and adolescents who actively participated in a fully enclosed WL summer camp that was not free and lasted for at least 3 weeks, so the subjects of this study might not be representative of all obese children and adolescents. Third, dietary habits may be the determinants of metabolic differences between MHO and MUO populations. Therefore, the neglect of change of dietary habits is one of the important shortcomings of this study. Future researchers should pay more attention to the impact of dietary habit changes on weight loss and metabolic changes in MHO/MUO children and adolescents. Additionally, the scattered abnormal changes in BP and FBG in the MHO group after intervention were likely attributed to measurement or operation errors, which should be take care of in future studies. Keeping subjects still before measurement of blood pressure may improve the accuracy of measurement. Finally, because metabolically healthy status in obese children and adolescents is susceptible with high plasticity, further long-term clinical randomized controlled trials are required to obtain more convincing time-to-event results.

## 5. Conclusions

In summary, our study shows that both MHO and MUO children and adolescents can benefit equivalently from diet- and exercise-based interventions in improvement of BW, BMI, BFR, WC, TC, and LDL-C levels, while SBP, DBP, RHR, FBG, and TG levels improved more obviously in the MUO group than in the MHO group. Thus, our findings support the opinion of highlighting the importance of metabolically healthy maintenance across all BMI groups among Chinese children and adolescents. Importantly, our results indicate that metabolically healthy status is transient and likely to modify during childhood and adolescence, a period which has vital indicative implications for maintaining a metabolically healthy state in adulthood. Early targeted interventions, such as strategies aiming at lowering BP and preventing the decrease in HDL-C level, should be considered for the precise treatment of obesity in clinical practice with the goal of improving metabolically healthy state.

## Figures and Tables

**Table 1 ijerph-19-06120-t001:** Basic characteristics of subjects who met the MHO/MUO criteria for children and adolescents in this study.

Variable	MHO	MUO	Total	*p* Value
N (%)	102 (36.2%)	180 (63.8%)	282 (100%)	/
Age (y)	12.4 ± 2.3	13.2 ± 2.2	12.9 ± 2.3	0.010
Sex (M/F)	49/53	102/78	151/131	0.163
Height (cm)	159.7 ± 11.5	163.9 ± 10.4	162.4 ± 11.0	0.002
BW (kg)	75.2 ± 18.1	85.3 ± 19.3	81.7 ± 19.4	<0.001
BMI (kg/m^2^)	29.0 ± 3.8	31.4 ± 4.6	30.5 ± 4.5	<0.001
WC (cm)	93.8 ± 11.1	99.4 ± 11.1	97.4 ± 11.4	<0.001

Data are presented as mean ± SD. *p* value means comparison of variables between MHO and MUO group. N, number; M/F, male/female; BW, body weight; WC, waist circumference.

**Table 2 ijerph-19-06120-t002:** Distribution of metabolically healthy indicators in MHO and MUO groups before and after intervention.

Variable	MHO Group (*n* = 102)		MUO Group (*n* = 180)	
	Before (%)	After (%)	Before (%)	After (%)
MUO status	0 (0.0%)	36 (35.3%)	180 (100%)	112 (62.2%)
BMI > 95th percentile ^1^	102 (100%)	94 (92.2%)	180 (100%)	164 (91.1%)
HDL-C ≤ 1.03 mmol/L	0 (0.0%)	29 (28.4%)	76 (42.2%)	102 (56.7%)
TG > 1.7 mmol/L	0 (0.0%)	0 (0.0%)	22 (12.2%)	2 (1.1%)
SBP > 90th percentile ^1^	0 (0.0%)	7 (6.9%)	117 (65.0%)	36 (20.0%)
DBP > 90th percentile ^1^	0 (0.0%)	6 (5.9%)	88 (48.9%)	32 (17.8%)
FBG > 5.6 mmol/L	0 (0.0%)	1 (1.0%)	3 (1.7%)	0 (0.0%)

^1^ Age- and sex-specific percentiles, for blood pressure, also height-specific. MHO, metabolically healthy obesity; MUO, metabolically unhealthy obesity; WL, weight loss; BMI, body mass index; HDL-C, high-density lipoprotein cholesterol; TG, triglycerides; SBP, systolic blood pressure; DBP, diastolic blood pressure; FBG, fasting blood glucose.

**Table 3 ijerph-19-06120-t003:** Changes of anthropometry and blood indicators in MHO and MUO groups before and after intervention.

Variable	MHO			MUO			*p* Value
	Before	After	Change% (95% CI)	Before	After	Change% (95% CI)	
BW (kg)	75.2 ± 18.1	68.3 ± 16.5 ^†,^**	−9.2 (−9.5, −8.9)	85.3 ± 19.3	77.3 ± 17.6 ^§,^**	−9.5 (−9.8, −9.2)	0.317
BMI (kg/m^2^)	29.0 ± 3.8	26.3 ± 3.5 ^†,^**	−9.3 (−9.6, −9.0)	31.4 ± 4.6	28.4 ± 4.3 ^§,^**	−9.7 (−10.0, −9.4)	0.077
BFR (%)	40.0 ± 4.7	35.6 ± 5.6 ^†,^**	−11.1 (−12.7, −9.6)	40.2 ± 5.5	35.5 ± 6.5 ^§,^**	−11.9 (−13.1, −10.7)	0.292
WC (cm)	93.8 ± 11.1	84.4 ± 10.4 ^†,^**	−9.9 (−11.0, −8.9)	99.4 ± 11.1	89.8 ± 10.6 ^§,^**	−9.56 (−10.3, −8.8)	0.357
SBP (mmHg)	107.5 ± 7.9	103.0 ± 9.5 ^†,^**	−3.9 (−5.8, −1.9)	124.8 ± 14.9	111.3 ± 13.5 ^§,^**	−10.2 (−11.8, 8.6)	<0.001
DBP (mmHg)	64.7 ± 6.4	62.3 ± 9.4 ^†,^**	−2.9 (−6.2, 0.4)	77.6 ± 14.8	65.7 ± 11.8 ^§,^**	−13.5 (−16.1, −10.8)	<0.001
RHR (beats/min)	88.9 ± 10.6	81.1 ± 11.9 ^†,^**	−8.3 (−10.7, −6.0)	93.1 ± 13.7	81.6 ± 13.1 ^§,^**	−11.1 (−13.6, −8.7)	0.025
FBG (mmol/L)	4.6 ± 0.4	4.5 ± 0.4	−0.5 (−2.4, 1.4)	4.6 ± 0.4	4.4 ± 0.4 ^§,^**	−3.2 (−4.6, −1.8)	0.011
TG (mmol/L)	0.8 ± 0.2	0.7 ± 0.2 ^†,^**	−8.1 (−12.3, −3.9)	1.1 ± 0.6	0.8 ± 0.3 ^§,^**	−18.4 (−22.4, −14.5)	<0.001
TC (mmol/L)	4.4 ± 0.7	3.5 ± 0.6 ^†,^**	−19.4 (−21.5, −17.3)	4.5 ± 1.0	3.5 ± 0.6 ^§,^**	−20.0 (−21.6, −18.4)	0.670
HDL-C (mmol/L)	1.3 ± 0.2	1.2 ± 0.2 ^†,^**	−8.6 (−10.8, −6.3)	1.1 ± 0.3	1.1 ± 0.2 ^§,^**	−6.2 (−8.0, −4.4)	0.121
LDL-C (mmol/L)	2.6 ± 0.6	0.4 ± 0.6 ^†,^**	−26.1 (−28.7, −23.6)	2.9 ± 0.7	1.0 ± 0.8 ^§,^**	−26.7 (−28.5, −24.8)	0.730

Data presented as the group means ± SD. ** *p* < 0.01. “^†^” and “^§^” mean comparison between before- and after-intervention in MHO group and MUO group, respectively. *p* value means comparison of percent changes in variables between MHO and MUO group. Baseline levels of dependent variables were adjusted for each ANCOVA analysis. For analysis of BW and blood pressure, age, sex, and height were also adjusted. For analysis of BMI, age and sex were also adjusted. For analysis of WC, age, sex, and BMI were also adjusted. BW, body weight; BMI, body mass index; BFR, body fat ratio; WC, waist circumference; SBP, systolic blood pressure; DBP, diastolic blood pressure; RHR, resting heart rate; FBG, fasting blood glucose; TG, triglycerides; TC, total cholesterol; HDL-C, high-density lipoprotein–cholesterol; LDL-C, low-density lipoprotein–cholesterol.

## Data Availability

Data sharing is not applicable to this article.

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
