# Peer review of "Effect of Diet and Exercise-Induced Weight Loss among Metabolically Healthy and Metabolically Unhealthy Obese Children and Adolescents"

_ijerph, 2022, doi:10.3390/ijerph19106120_

Round 1

Reviewer 1 Report

Language and technical care:

The manuscript requires some attention in terms of the following language and technical aspects:

  • Page 1, line 42 – a space between (CVC) and [1,2];
  • Page 2, line 55 to 57 – this entire sentence needs clarification or rephrasing;
  • Page 2, line 61 – perhaps the words ‘will contribute’ should rather be ‘can be used to contribute’;
  • Page 2, line 66 & 67 – insert the words ‘in the’ and ‘at the’ so that the sentence reads: “…who volunteered to participate in the WL Summer Camp program at the Shanghai Dian…”;
  • Page 2, line 72 – change to ‘examinations’ and ‘programs’;
  • Page 2, line 78 to 84 – the entire sentence needs clarification or rephrasing;
  • Page 3, line 112 – insert the word ‘The’ before Exercise program;
  • Page 3, line 119 – remove ‘a’, and insert ‘s’ – “…develop good lifestyle habits…”;
  • Page 3, line 123 – add a ‘s’ to ‘surveys’;
  • Page 3, line 140 – insert ‘a’ and change ‘was’ to ‘is’, as the definition still exists;
  • Page 4, line 170 – this reviewer is unsure about the word “Generally” in context of these results;
  • Page 7, line 232 – ‘obesity’ should be ‘obese’;
  • Page 7, line 236 – add the word ‘the’, so the sentence reads “…between the two groups…”;
  • Page 8, line 324 – consider replacing the word ‘quiet’ with ‘still’ – quiet sounds as if they should not make any sounds, while the object may have been to keep them from moving too much;
  • Page 9, line 334 – either ‘opinions’ or ‘the opinion’;

The manuscript is very well referenced using relevant, up-to-date references.

Literature Review:

This reviewer identified no shortcomings in the literature review – it is up to date and comprehensive, and contextualises this work well with other similar dietary nutrition research.

Methodology and materials:

The reviewer believes that the methodology is straightforward and explained well, and that good and acceptable research procedures have been followed. If a little more specific detail of the content of the intervention was provided, it might make it easier for the reader to understand what was done.

Results and Discussion:

The reviewer believes that the study results are of high quality and value, particularly in context of the growing nature of eating behaviour and consequent health as a result of food choices and lifestyles amongst children and adolescents.

Conclusion:

The limitations offered in the work is comprehensive and shows that the researchers are fully aware of the shortcomings of the work.

The reviewer believes that the conclusion is well written and the worthiness of the research is evident, particularly in light of our never-ending fight against the global onslaught of diet-related illnesses amongst those still at school.

Author Response

Comments and Suggestions for Authors

Language and technical care:

The manuscript requires some attention in terms of the following language and technical aspects.

Response: Thank you for pointing out those issues/oversights. We have rephrased or corrected those points in the revised manuscript.

The manuscript is very well referenced using relevant, up-to-date references.

Response: Thank you for this positive comment.

Literature Review:

This reviewer identified no shortcomings in the literature review – it is up to date and comprehensive, and contextualises this work well with other similar dietary nutrition research.

Response: Thank you for this positive comment.

Methodology and materials:

The reviewer believes that the methodology is straightforward and explained well, and that good and acceptable research procedures have been followed. If a little more specific detail of the content of the intervention was provided, it might make it easier for the reader to understand what was done.

Response: Thank you. More specific detail of the content of the intervention, including types of food and the distribution of training loads or exercises, has been added to the “Material and Methods” section. 

Results and Discussion:

The reviewer believes that the study results are of high quality and value, particularly in context of the growing nature of eating behaviour and consequent health as a result of food choices and lifestyles amongst children and adolescents.

Conclusion:

The limitations offered in the work is comprehensive and shows that the researchers are fully aware of the shortcomings of the work.

The reviewer believes that the conclusion is well written and the worthiness of the research is evident, particularly in light of our never-ending fight against the global onslaught of diet-related illnesses amongst those still at school.

Response: Thank you for these positive comments.

Reviewer 2 Report

- The introduction can be improved, it is good, however, it can be deepened to give greater consistency to the discussion of the study.

- In the introduccion, what do yopu mean by traditional treatment?.

- Informed assent must be incorporated in material and methods.

- The methodology of the field work should be described more clearly, that is, in addition to what is indicated, state the distribution of training loads orexercises. It is necessary to express a session type.

- The article should always be written in the third person.

Author Response

Comments and Suggestions for Authors

- The introduction can be improved, it is good, however, it can be deepened to give greater consistency to the discussion of the study.

Response: Thank you for the comment. We have deepened the “Introduction” section as suggested. We have added: (1) a description of the clinical features of obesity in children and adolescents, and (2) current research progress on short- and long-term metabolism and lifestyle changes after diet- and exercise-based weight loss in obese children and adolescents in the “Introduction” section.

- In the introduction, what do you mean by traditional treatment?

Response: Thank you for this comment. We have detailed the content of the traditional diet- and exercise-based intervention that are often used in China for weight management, including types of food and the distribution of training loads or exercises, in the “Material and Methods” section.

- Informed assent must be incorporated in material and methods.

Response: Informed assent has been incorporated in the “2.1. Study Population and Database” section in the 1st submission.

- The methodology of the field work should be described more clearly, that is, in addition to what is indicated, state the distribution of training loads or exercises. It is necessary to express a session type.

Response: Thank you for this suggestion. More specific detail of the content of the intervention, including types of food and the distribution of training loads or exercises, has been added to the “Material and Methods” section.

- The article should always be written in the third person.

Response: Thank for this suggestion. The entire text has been revised to third-person writing mode.

Reviewer 3 Report

This research documented the results of a 3 to 4-week diet and exercise based intervention on weight loss and cardio-metabolic health state in a sample of obese children and adolescents, differentiated between metabolic healthy obese (MHO) and metabolic unhealthy obese (MUO). Authors evaluated (before and after intervention) anthropometric variables, blood pressure, resting heart rate, triglycerides, fasting blood glucose, total cholesterol, low-density lipoprotein-cholesterol and high-density lipoprotein-cholesterol levels.

Although the research, data analysis and interpretation are scientifically valid and well-detailed, some important requests concerning an intervention on data are essential, as an analysis divided by sex and age is missing.

The problem of obesity in these age ranges is not fully described in the Introduction as it does not take into account sex and age factors in the State of Art description. Moreover, citations and specific examples of analogues interventions which detailed the effectiveness (or not) of such interventions in the short or long period are also missing. Concerning this topic, I think it is really important (and well-documented in Literature) to highlight not only the effectiveness at the end of the intervention but also the long-term effects (in a follow-up) to verify the improvement of lifestyle behaviour. It should be necessary to explore this aspect and document it either in the literature analysis in relation to the age ranges, or in the discussion as well as in the future addressing of this research.

Below the details of intervention and modification:

Rows 125-127: the sequence of height and BW should be reversed as “BW and height were measured…”

Rows 128-129: why did you “estimate” waist circumference instead of measuring it with a medical tape?

Row 132: It should be necessary to explicit the reference concerning the manufacturer’s instructions.

From row 150 to row 161 of Material and Methods: the analyses are well-described and detailed, although I have some concerns:

  • Why didn’t you divide the sample by sex and age? Some anthropometric and metabolic variables are very different depending on these factors and maybe you could detect more in-depth changes with such approach.
  • The age range (6-19 years, described in the exclusion criteria at row 79) is quite big: I suggest to perform analyses by also dividing the sample by age ranges based on the references that you consider appropriate.

Rows 185-186 and 288:  in concern to “data not shown”, could you present such data in a supplementary file?

Results:

  • In the tables, please uniform the way of indicating statistical significance.

It is expectable that differentiating analyses by sex and age, the effectiveness of the suggested intervention could be potentially higher for some subgroups so that it will be possible to address future interventions which can improve health conditions. With such approach you could maybe find more in-depth interpretation suggestions also in regard to HDL-C results.

Author Response

Comments and Suggestions for Authors

This research documented the results of a 3 to 4-week diet and exercise based intervention on weight loss and cardio-metabolic health state in a sample of obese children and adolescents, differentiated between metabolic healthy obese (MHO) and metabolic unhealthy obese (MUO). Authors evaluated (before and after intervention) anthropometric variables, blood pressure, resting heart rate, triglycerides, fasting blood glucose, total cholesterol, low-density lipoprotein-cholesterol and high-density lipoprotein-cholesterol levels.

Although the research, data analysis and interpretation are scientifically valid and well-detailed, some important requests concerning an intervention on data are essential, as an analysis divided by sex and age is missing.

The problem of obesity in these age ranges is not fully described in the Introduction as it does not take into account sex and age factors in the State of Art description. Moreover, citations and specific examples of analogues interventions which detailed the effectiveness (or not) of such interventions in the short or long period are also missing. Concerning this topic, I think it is really important (and well-documented in Literature) to highlight not only the effectiveness at the end of the intervention but also the long-term effects (in a follow-up) to verify the improvement of lifestyle behaviour. It should be necessary to explore this aspect and document it either in the literature analysis in relation to the age ranges, or in the discussion as well as in the future addressing of this research.

Response: Thank you for the comment. As suggested, we have examined whether the observed findings differed by sex (male and female) or age groups (6-12 and 13-19 years) by including interaction terms of sex * groups (MUO vs. MHO) and age * groups (MUO vs. MHO). We found no evidence of sex or age interactions indicating the findings were similar between sex or age groups. To maximize the power, we decided to conduct the analysis in the whole sample. We have also added one sentence regarding the interaction analysis in the “Statistical analyses”- “Interactions of sex and age group with groups (MUO and MHO) were also considered. As no evidence of interactions were observed. The analysis was conducted using the whole sample”.

In the “Introduction” section, we have also added: (1) a description of the clinical features of obesity in children and adolescents, and (2) current research progress on short- and long-term metabolism and lifestyle changes after diet- and exercise-based weight loss in obese children and adolescents. In addition, we have highlighted in the “Discussion” section that future attention should be paid to age-and sex-based metabolic changes in children and adolescents after weight loss.

Below the details of intervention and modification:

Rows 125-127: the sequence of height and BW should be reversed as “BW and height were measured…”

Response: Thank you for this suggestion. We have updated this sentence as suggested.

Rows 128-129: why did you “estimate” waist circumference instead of measuring it with a medical tape?

Response: Thank you for pointing out this issue. The term of “estimate” is misused here and we have changed it to “measured”.

Row 132: It should be necessary to explicit the reference concerning the manufacturer’s instructions.

Response: Thank you for this comment. We have added related reference about manufacturer’s instructions where needed. 

From row 150 to row 161 of Material and Methods: the analyses are well-described and detailed, although I have some concerns:

Why didn’t you divide the sample by sex and age? Some anthropometric and metabolic variables are very different depending on these factors and maybe you could detect more in-depth changes with such approach.

The age range (6-19 years, described in the exclusion criteria at row 79) is quite big: I suggest to perform analyses by also dividing the sample by age ranges based on the references that you consider appropriate.

Response: Thank you for these comments. For obese children and adolescents, analysis based on age and gender is critical, as body morphological development during this period varies greatly by age and gender. However, this study focused on the MHO/MUO population of children and adolescents. In the components of MHO consensus criteria, both obesity and abnormal blood pressure are stratified by age and sex. In addition, we also adjusted for confounding factors that might affect the results when performing data analysis. Therefore, we have considered the possible influence of age and gender factors. The pre- and post-intervention indicators shown in Table 3, such as the values of BMI and other indicators, represent the mean values of MHO/MUO children and adolescents, rather than the obese children and adolescents groups that we usually refer to. We also found some analogous studies on childhood and adolescent obesity exhibited a similar pattern of presentation of results (with large age ranges, but showed mean values for the groups) [1-3].

Rows 185-186 and 288:  in concern to “data not shown”, could you present such data in a supplementary file?

Response: Thank you for this suggestion. We have provided a supplementary file in the revised version.

Results:

In the tables, please uniform the way of indicating statistical significance.

Response: Thank for this suggestion. we have checked to ensure the statistical significance is presented in a uniform manner in the tables.

It is expectable that differentiating analyses by sex and age, the effectiveness of the suggested intervention could be potentially higher for some subgroups so that it will be possible to address future interventions which can improve health conditions. With such approach you could maybe find more in-depth interpretation suggestions also in regard to HDL-C results.

Response: Thank you for this comment. As mentioned above, a description of the interaction analysis between age groups or sex with metabolic status groups are also added to the "2.5. Statistical Analyses" section.

References

  1. Dao, H.H.; Frelut, M.L.; Peres, G.; Bourgeois, P.; Navarro, J. Effects of a multidisciplinary weight loss intervention on anaerobic and aerobic aptitudes in severely obese adolescents. Int J Obes Relat Metab Disord 2004, 28, 870-878, doi:10.1038/sj.ijo.0802535.
  2. Genovesi, S.; Antolini, L.; Orlando, A.; Gilardini, L.; Bertoli, S.; Giussani, M.; Invitti, C.; Nava, E.; Battaglino, M.G.; Leone, A.; et al. Cardiovascular Risk Factors Associated With the Metabolically Healthy Obese (MHO) Phenotype Compared to the Metabolically Unhealthy Obese (MUO) Phenotype in Children. Front Endocrinol (Lausanne) 2020, 11, 27, doi:10.3389/fendo.2020.00027.
  3. Prince, R.L.; Kuk, J.L.; Ambler, K.A.; Dhaliwal, J.; Ball, G.D. Predictors of metabolically healthy obesity in children. Diabetes Care 2014, 37, 1462-1468, doi:10.2337/dc13-1697.

Reviewer 4 Report

  1. Line 17 and throughout - I think it should be metabolically healthy/unhealthy for grammatical sense rather than metabolic. Please check and update throughout the manuscript.
  2. Line 50 - should this read "confirmed" rather than "conformed"?
  3. Line 58 and 229 - the authors describe the dietary intervention as "traditional". It may be more accurate to describe this as culturally-relevant and/or aligned with national food-based guidelines.Without knowledge of what types of food were provided, it is difficult to know if the intervention was "traditional". The mode of intervention (holiday camps delivering personalised recommendations) probably can't be defined as being traditional though#. Consider rephrasing in a more appropriate and less-ambiguous fashion.
  4. Table 1 - consider whether presentation of absolute BMI values is useful as a characteristic for a child group of mixed age range here, particularly as criteria for inclusion relate to BMI by age. I'm also aware there are limitations in the use of waist circumference in younger groups of children, which I think would include some of the participants of this study. Some groups suggest using different cut-points for different age groups . Consider the overall relevance of body measurements provided here and whether this could be presented in a way that summarises the study cohort more rationally.
  5. Table 3 - age-adjusted BMI may make more sense to summarise in this Table. While the intervention is relatively brief, there is still potential that a proportion of children have had a birthday during the intervention and potentially would need BMI data considered separately. Collation of absolute BMI and waist circumference values may not be sensible here. As before and after (paired tests) have been carried out, would it be more useful to present mean changes in parameters for participants (e.g. mean [%?] weight lost/ mean improvement in biochemical parameters? That is what the test is making a comparison of rather than mean weight at time point A for the cohort versus mean weight at timepoint B for the cohort?
  6. Line 232 - update to "less obese". I note that you would probably need to show the proportion of children now classified as being in the normal weight/overweight categories for their age to be able to make this statement. "Of lower excess body weight" may otherwise be more appropriate.
  7. General - based on suggestions for updates to the Results section, there may be a need for further updates in the Discussion to align appropriately. As a broad comment, one major limitation would be not having information on dietary habit change before the intervention. Presumably this is available during the intervention (see also comment below)?
  8. General - Do you have evidence that participants complied with all targeted elements of their intervention (both dietary and physical activity)?

Author Response

Comments and Suggestions for Authors

Line 17 and throughout - I think it should be metabolically healthy/unhealthy for grammatical sense rather than metabolic. Please check and update throughout the manuscript.

Response: Thank you for this suggestion. We have updated the full text where the phrase "metabolically healthy" needs to be used.

Line 50 - should this read "confirmed" rather than "conformed"?

Response: Thank you for pointing out this oversight. We have corrected this error.

Line 58 and 229 - the authors describe the dietary intervention as "traditional". It may be more accurate to describe this as culturally-relevant and/or aligned with national food-based guidelines. Without knowledge of what types of food were provided, it is difficult to know if the intervention was "traditional". The mode of intervention (holiday camps delivering personalised recommendations) probably can't be defined as being traditional though#. Consider rephrasing in a more appropriate and less-ambiguous fashion.

Response: Thank you for the comment. More specific detail of the content of the intervention, including types of food and the distribution of training loads or exercises, has been added to the “2.2. Diet and Exercise Protocol” section.

Table 1 - consider whether presentation of absolute BMI values is useful as a characteristic for a child group of mixed age range here, particularly as criteria for inclusion relate to BMI by age. I'm also aware there are limitations in the use of waist circumference in younger groups of children, which I think would include some of the participants of this study. Some groups suggest using different cut-points for different age groups. Consider the overall relevance of body measurements provided here and whether this could be presented in a way that summarises the study cohort more rationally.

Response: Thank you for your comment. This is a very worthwhile question. As you said, absolute BMI values stratified by age and sex do not represent actual BMI levels for obese children and adolescents. In fact, evaluation of childhood and adolescent obesity is usually stratified by age and sex, making it difficult to accurately assess the actual BMI level of this groups. In Table 1, we tended to present the anthropometric characteristics of groups who meet the MHO/MUO criteria for children and adolescents. Excessive age dispersion will have an impact on the accuracy and representativeness of the results presented. We have added special instructions in the header of table 1 and highlighted concerns about this issue in the limitations part of the “Discussion” section. In addition, we reviewed analogous studies [2,3] and found that the basic characteristics of MHO/MUO children and adolescents were described in a similar way to this study.

Table 3 - age-adjusted BMI may make more sense to summarise in this Table. While the intervention is relatively brief, there is still potential that a proportion of children have had a birthday during the intervention and potentially would need BMI data considered separately. Collation of absolute BMI and waist circumference values may not be sensible here. As before and after (paired tests) have been carried out, would it be more useful to present mean changes in parameters for participants (e.g. mean [%?] weight lost/ mean improvement in biochemical parameters? That is what the test is making a comparison of rather than mean weight at time point A for the cohort versus mean weight at timepoint B for the cohort?

Response: Thank you for your comment. For analysis of BMI in table 3, age and sex were both adjusted to avoid the confounding effects due to duration of intervention. Please check the footnotes of table 3. In this study, we focused the specific population MHO/MUO children and adolescents. In the childhood and adolescent MHO/MUO criteria, obesity and abnormal blood pressure are already defined referring to age- and sex-matched standard body weight. Therefore, the absolute values of BMI and waist circumference before and after the intervention represent the changes of the specific MHO/MUO group rather than the characteristics of what we usually call obese children and adolescents. In table 3, we show the mean of pre- and post-intervention and the mean of % change, which we think may be appropriate to display the magnitude of the change in these indicators. In addition, we have examined whether the observed findings differed by sex (male and female) or age groups (6-12 and 13-19 years) by including interaction terms of sex * groups (MUO vs. MHO) and age * groups (MUO vs. MHO). We found no evidence of sex or age interactions indicating the findings were similar between sex or age groups. To maximize the power, we decided to conduct the analysis in the whole sample.

Line 232 - update to "less obese". I note that you would probably need to show the proportion of children now classified as being in the normal weight/overweight categories for their age to be able to make this statement. "Of lower excess body weight" may otherwise be more appropriate.

Response: Thank you. We have used the phrase of “of lower excess body weight” as suggested.

General - based on suggestions for updates to the Results section, there may be a need for further updates in the Discussion to align appropriately. As a broad comment, one major limitation would be not having information on dietary habit change before the intervention. Presumably this is available during the intervention (see also comment below)?

Response: Thank you for the comment. It is true that diet is a critical component of intervention programs and has a direct impact on intervention outcomes. This study mainly focused on the differences in the weight loss effects of diet and exercise interventions in obese groups with different metabolic states, thus, ignoring the attention to changes in eating habits is one of the shortcomings of this study. We have addressed this in the "limitations" paragraph of the Discussion section, and suggest that future researchers focus on the impact of dietary habit changes on weight loss in the MHO/MUO population of children and adolescents. Nonetheless, in this study, we were able to ensure that the dietary program was effectively implemented during the intervention process, because we adopted a closed weight management model, and professional nutritionists will provide personalized meals to each subject. We detailed the main elements of the dietary intervention in the “Methods and Materials” section to make it easier for readers to understand the dietary intervention program.

General - Do you have evidence that participants complied with all targeted elements of their intervention (both dietary and physical activity)?

Response: Thank you for this comment First, we adopted a fully closed weight loss management model. Eating in designated dining halls and living in strictly managed dormitories would largely avoid other food sources. Second, we assigned a teaching assistant to each coach in the class to monitor the physical activity of the subjects during the class. Therefore, we believe that these measures can ensure that participants complied with targeted elements of their interventions to a large extent.

Round 2

Reviewer 3 Report

The authors have fulfilled all the requested revisions.